# A Novel Precision Approach to Overcome the “Addiction Pandemic” by Incorporating Genetic Addiction Risk Severity (GARS) and Dopamine Homeostasis Restoration

**DOI:** 10.3390/jpm11030212

**Published:** 2021-03-16

**Authors:** Kenneth Blum, Shan Kazmi, Edward J. Modestino, Bill William Downs, Debasis Bagchi, David Baron, Thomas McLaughlin, Richard Green, Rehan Jalali, Panayotis K. Thanos, Igor Elman, Rajendra D. Badgaiyan, Abdalla Bowirrat, Mark S. Gold

**Affiliations:** 1College of Osteopathic Medicine of the Pacific, Western University of Health Sciences, Pomona, CA 91766, USA; shan.kazmi@westernu.edu (S.K.); dbaron@westernu.edu (D.B.); 2Institute of Psychology, ELTE Eötvös Loránd University, 1117 Budapest, Hungary; 3Division of Nutrigenomics, The Kenneth Blum Behavioral Neurogenetic Institute, Austin, TX 78712, USA; tmclaugh50@gmail.com (T.M.); rickgreen@newresourcesmedicalarts.com (R.G.); rjalai@ivitalize.com (R.J.); 4Department of Psychiatry, University of Vermont, Burlington, VT 05405, USA; 5Department of Psychiatry, Wright University Boonshoff School of Medicine, Dayton, OH 45435, USA; 6Division of Precision Nutrition, Victory Nutrition International, Lederach, PA 19450, USA; billd@vni.life (B.W.D.); debasisbagchi@gmail.com (D.B.); 7Center for Genomic Testing, Geneus Health LLC, San Antonio, TX 78249, USA; 8Department of Psychology, Curry College, Milton, MA 02186, USA; edward.modestino@gmail.com; 9Department of Pharmaceutical Sciences, College of Pharmacy & Health Sciences, Texas Southern University, Houston, TX 77004, USA; 10Precision Translational Medicine (Division of Ivitalize), San Antonio, TX 78249, USA; 11Department of Psychology & Behavioral Neuropharmacology and Neuroimaging Laboratory on Addictions (BNNLA), Research Institute on Addictions, University at Buffalo, Buffalo, NY 14260, USA; pkthanos@gmail.com; 12Department of Psychiatry, Harvard University, School of Medicine, Cambridge, MA 02142, USA; dr.igorelman@gmail.com; 13Department of Psychiatry, South Texas Veteran Health Care System, Audie L. Murphy Memorial VA Hospital and Long School of Medicine, University of Texas Health Science Center, San Antonio, TX 78249, USA; badgaiyan@gmail.com; 14Department of Psychiatry, MT. Sinai School of Medicine, New York, NY 10003, USA; 15Department of Molecular Biology and Adelson School of Medicine, Ariel University, Ariel 40700, Israel; bowirrat@gmail.com; 16Department of Psychiatry, Washington University School of Medicine, St. Louis, MO 63110, USA; drmarkgold@gmail.com

**Keywords:** enkephalinase-Inhibition, hypodopaminergia, Reward Deficiency Syndrome (RDS), dopamine homeostasis, Pro-dopamine regulation, Genetic Addiction Risk System (GARS)

## Abstract

This article describes a unique therapeutic precision intervention, a formulation of enkephalinase inhibitors, enkephalin, and dopamine-releasing neuronutrients, to induce dopamine homeostasis for detoxification and treatment of individuals genetically predisposed to developing reward deficiency syndrome (RDS). The formulations are based on the results of the addiction risk severity (GARS) test. Based on both neurogenetic and epigenetic evidence, the test evaluates the presence of reward genes and risk alleles. Existing evidence demonstrates that the novel genetic risk testing system can successfully stratify the potential for developing opioid use disorder (OUD) related risks or before initiating opioid analgesic therapy and RDS risk for people in recovery. In the case of opioid use disorders, long-term maintenance agonist treatments like methadone and buprenorphine may create RDS, or RDS may have been in existence, but not recognized. The test will also assess the potential for benefit from medication-assisted treatment with dopamine augmentation. RDS methodology holds a strong promise for reducing the burden of addictive disorders for individuals, their families, and society as a whole by guiding the restoration of dopamine homeostasisthrough anti-reward allostatic neuroadaptations. WC 175.

## 1. Introduction

We are facing an incredible challenge in combatting the current opioid and drug pandemic worldwide. Although there has been notable progress, in 2017 alone, 72,000 people died from a narcotic overdose in the USA. The National Institute on Alcohol Abuse and Alcoholism (NIAAA) and National Institute on Drug Abuse (NIDA) continue to struggle with cultivating innovations to impede or eliminate this unwanted epidemic. The FDA list of approved medication assistance treatment (MAT) works primarily by blocking dopamine release and function at the pre-neuron in the nucleus accumbens [1,2]. Although MAT have reduced overdose deaths, health care events, and costs, a long-term strategy to return MAT patients to premorbid functioning is needed. Medication-assisted treatments often fail [3], and when discontinued, relapse and overdose occur at high rates similar to those of untreated patients. Neurologically, MATs may induce persistent changes that compromise endorphin, dopamine, and other brain systems. Long-term agonist treatments may be necessary for lack of other options, but we caution that data on long-term use vs. short-term harm reduction is lacking [4]. 

During the current viral pandemic, there is also a profound, pre-existing, and growing worldwide epidemic of addiction to opioids, psychostimulants, alcohol, and cannabis. Treatments themselves, like long-term agonist treatments for opioid use disorder (OUD), may also cause Reward Deficiency Syndrome (RDS) [4], resulting in harm and deadly consequences that rival and might eclipse the magnitude of the current viral issue. 

The devastation and number of deaths due to drug overdose, while highest in the United States, is a global issue requiring “out of the box” thinking. The short-term consequences of opioid substitution therapy can be a reduction of harm. The long-term consequences can lock people into unwanted and potentially lethal addictions [5]. Attempting to curb the usage of opioids by giving potent opioids seems fundamentally incongruous. An alternative approach is to use the narcotic antagonist (like naltrexone) to induce “psychological extinction” (weakening of a conditioned response over time) by blocking delta and Mu opioid receptors [6]. This latter approach using narcotic antagonism seems to be more acceptable, but compliance is an issue, moderated by the patient’s genetic antecedents [7]. There are other approaches with FDA approval for alcoholism, but they also seem to block dopaminergic signaling [8,9]. 

### The Reward Deficiency Syndrome (RDS) Concept

Understanding the above premise and emerging acceptance of the underlying concept of reward deficiency syndrome (RDS), which Blum first conceived of in 1995, facilitates the common mechanism hypothesis for drug and behavioral addictions [10]. The common neuromodulating aspects of neurotransmission and its disruption from chronic exposure to drugs and behavioral addictions necessitates an approach involving the attainment of “dopamine homeostasis” [11].

## 2. Snapshot of Evidence: Biochemical and Genetic Dysfunctions That Are Evident in the Context of RDS

This “out of the box” novel approach requires the use of genetic risk polymorphic testing together with a safe and well-researched neuronutrient KB220z, customized to match the hypodopaminergic risk alleles identified in the individuals’ genetic test. The presented evidence-validated precision nutrigenomic technology is known to have pro-dopamine regulatory pharmacological properties [12]. 

Several lines of evidence support a common neural mechanism between substance and non-substance addiction (like alcohol, opioids, food) [13,14,15,16]. In the 1970s, Blum’s laboratory developed an amino-acid based, enkephalinase-inhibitory, pro-dopamine regulator (PDR) into the KB220 nutraceutical, which now has improved variants and over 45 published clinical studies [17]. The basis of this complex is that it mimics and rebalances the gene expression in the brain reward cascade (BRC), an established model of reward processing. The most striking feature of KB200Z therapy is the normalization of motivational and brain reward function, improving overall functional competence. The areas involved include the nucleus accumbens, anterior cingulate gyrus, anterior thalamic nuclei, hippocampus, and prelimbic/infralimbic loci in animals and abstinent heroin addicts [18,19]. It is noteworthy that the synergistic effect of N-Acetyl cysteine, an ingredient necessary for glutathione synthesis included in the KB220 variants, may help facilitate the improvement of functional connectivity [20]. Notably, there is increasing evidence that redox dysregulation, which can lead to oxidative stress and impairment of oligodendrocytes and parvalbumin interneurons, may underlie brain connectivity alterations in schizophrenia. Increased brain antioxidant glutathione levels in the medial prefrontal cortex correlated positively with increased functional connectivity within the cingulum bundle in healthy controls; this was not the case for early psychosis patients. In a recent randomized controlled trial, Mullier et al. [21] observed that 6-month supplementation with a glutathione precursor, specifically N-acetyl cysteine, increased brain glutathione levels, improved symptomatic expression and processing speed. Moreover, these researchers found that N-acetyl-cysteine supplementation increases functional connectivity along the cingulum and, more precisely, between the caudal anterior part and the isthmus of the cingulate cortex. It seems sensible that by increasing glutathione and potentiating healthy oxygen utilization, N- acetylcysteine may help overcome certain RDS behaviors. 

## 3. Reward Deficiency Syndrome (RDS); A Behavioral Octopus 

Blum’s group has provided evidence related to many RDS behaviors by utilizing a glutaminergic–dopaminergic optimization complex over the past 50 years, and advance goals have forced variants of KB220. Specifically, the development of a glutaminergic–dopaminergic optimization complex, KB220Z, provides the brain reward systems with the potential to balance neurotransmission and initiate “dopamine homeostasis.” This is acquired by restoring optimal gene expression and rebalancing neurotransmitters and neural interconnectivity. The KB220Z has been the subject of many studies of all types, including but not limited to triple and double-blinded, placebo-controlled, and peer-reviewed articles [22]. This complex may provide substantial clinical benefit to the victims of reward deficiency syndrome (RDS) and help recovery from iatrogenically induced addiction to unwanted opiates/opioids and other addictive behaviors.

Individuals with a mood disorder, addiction, personality disorder, and impulsivity may share a dysfunction in how the brain perceives reward, particularly where the processing of natural endorphins or the response to exogenous dopamine stimulants is impaired. Reward deficiency syndrome (RDS) is a polygenic trait with implications that suggest impaired crosstalk between different neurological systems, including the known reward pathway, neuroendocrine systems, and motivational systems. Remarkably, substance use disorder (SUD), major depressive disorder (MDD), early life stress, immune dysregulation, attention deficit hyperactivity disorder (ADHD), post-traumatic stress disorder (PTSD), compulsive gambling, and compulsive eating disorders could be subtypes of overlapping, interrelated, neurochemical dysfunction. These disorders recruit underlying reward deficiency mechanisms in multiple brain centers. This array of associated and overlapping behavioral manifestations have hypodopaminergia in common, and the basic endophenotype recognized as RDS is likened to a behavioral octopus [13,14].

Drug, food, or behavioral addictions characterized by reduced dopaminergic function linked earlier to the DRD2 gene A1 allele (1), and now an array of reward gene alleles are manifest clinically as reward deficiency syndrome (RDS) [23]. Indeed, animal models of RDS are available [24]; in fact, Willuhn et al. [25] showed that there is a phasic dopamine deficiency as seen in RDS when animals increase lever pressing for more cocaine. There is also evidence for the potential utility of a natural, non-addictive, and safe putative D2 agonist that can treat patients in recovery from reward deficiency syndrome (RDS), including psychoactive substance use disorder (SUD) [26]. 

## 4. The Cascade of Neurotransmission—The Blueprint 

The brain reward cascade (BRC) schematic depicts the known interactions of at least seven neurotransmitter-pathways involved in the neurotransmission of reward. The complexity of the interactions within the BRC demand frequent revision of this schematic (see Figure 1). 

## 5. Measuring the Neurogenetics of Addiction Risk and Developing the GARS Test

In previous work, Blum’s group developed the genetic addiction risk severity (GARS) test following seminal research conducted by Blum’s group in 1990, which identified the first genetic association with severe alcoholism published in JAMA [27]. Although no one has provided sufficient RDS-free controls, and many of these so-called controls (e.g., blood donors) are disputed [28]. This lack of disease-free case controls remains in the field, and spurious results continue confusion regarding the role of genetics in addiction. Despite this conversely, many studies have used case controls that have primarily eliminated SUD. An estimation based on previous case-controlled studies available from the literature reveals significant associations between alcohol and substance risk alleles. Indeed, a total of 110,241 cases and 122,525 controls provides evidence that indicates an association of these risk alleles (measured in GARS), mostly indicating hypodopaminergia in the subjects. The instrumentation, data collection procedures, and the analytical approaches used to develop the GARS test have been published elsewhere (see [29]). 

The GARS-test screening will offer a novel opportunity to identify causal pathways, and mechanisms of psychological characteristics, genetic factors involved in addictions. Additional scientific evidence, including a future meta-analysis of all available data, is a work in progress.

## 6. Genetic Polymorphisms of RDS: Case-Control Studies, for Alcoholism

Next-generation large-scale genomics studies have had limited success in identifying alleles associated with addiction and RDS. Although Genome Wide Association Studies (GWAS) and next generation sequencing are now available as important genetic tools, there are some fundamental issues. Certainly, GWAS, for example, is useful to identify new clusters of genes that may relate to an etiological factor as a genetic antecedent to specific RDS behaviors like chemical dependency. The next important step following GWAS results is subsequent convergence to specific candidate genes. Thus, if there is indeed a blueprint or clue as to a specific known gene and associated polymorphic risk allele to link to a specific phenotype such as SUD or even cannabis use disorder, although the contribution of each gene may be small, it is still significant. 

Being cognizant of these difficulties and awaiting further research, the BRC was utilized as a blueprint, we reviewed the literature to determine each allele and associated polymorphism proposed in the GARS panel in case-control studies, specifically for alcoholism (see Table 1). 

## 7. Genetic Vulnerability: Clinical Implications 

Genetic vulnerability to unwanted addictive behavior can be identified early in life [30]. Based on a relatively moderate amount of published literature, reward gene polymorphisms predispose individuals to an increased risk of all subtypes of RDS behaviors, including anhedonia [10]. The genetic addiction risk score (GARS) test has been developed to identify one’s risk potential for these addictive-like behaviors. Specifically, published studies illustrate the GARS test’s use to identify specific neurotransmitter pathways where the risk for a signal breakdown in the BRC occurs and uses semi-customized precision KB220Z variants, matched the individuals’ GARS test result to treat the dysfunction. This synergistic ‘systems biology’ approach provides an increased efficacy in treating RDS [28,31,32,33]. Dopamine is a major neurotransmitter involved in substance and behavioral addictions; however, there is controversy about managing dopamine clinically to prevent and treat many addictive disorders. 

## 8. Induction of Dopamine Homeostasis

There is generally a consensus that balancing the brain reward circuit or achievement of “dopamine homeostasis” is a worthwhile goal, rather than blocking natural dopamine or administering a powerful opioid to manage opioid addiction [34]. We are inviting both the neuroscience, brain mapping, and clinical science communities to adopt this disruptive technology. With the future in mind, addressing this problem is by increasing worldwide research to explore these concepts to identify what constitutes “standard of care.” While harm reduction saves lives, the goal is to provide a path to a widely accepted new ‘standard of care’ for the induction of “dopamine Homeostasis.” This research goal is imperative in the face of our current psychostimulant, opioid, alcohol, and food addiction epidemic showing clinical relevance.

Due to the environment’s effect, all individuals are very unlikely to express all putative risk alleles. Based on our Quantitative Electroencephalogram (qEEG) studies and previous research [35], we cautiously state that long-term activation of dopaminergic receptors (i.e., DRD2 receptors) will give rise to a greater proliferation of the receptors and result in enhanced “dopamine sensitivity.” Dopamine receptor proliferation can lead to an increased sense of happiness, particularly in carriers of the DRD2 A1 allele [34]. Based on genetic and previous research [36], both treatment and prevention of multiple addictions, such as dependence on glucose, nicotine, and alcohol, could involve a biphasic approach. Thus, acute treatment could consist of medication-based preferential blocking of postsynaptic nucleus accumbens (NAc) opioid (delta, mu etc.) and dopamine receptors (D1–D5). However, both short term and long-term activation of the mesolimbic dopaminergic system using the KB220Z nutrigenomic technology can induce activation and release of dopamine (DA) at the NAc site to achieve DA homeostasis. An inability to effectively utilize either or both of these strategies and achieve DA homeostasis will result in continued abnormal behavior, mood, and potential suicidal ideation. Examples are those who possess a paucity of dopaminergic and serotonergic receptors or an increased rate of synaptic DA catabolism due to expression of the high catabolic genotype from the Catechol-O-methyltransferase(COMT) gene are predisposed to self-medicating via any substance or behavior such as alcohol, nicotine, psychostimulants, opiates, sex, excessive eating, gambling, and gaming, that will activate DA release. The high catabolic genotype from the COMT gene is the Val allele functions to catabolize dopamine in synapse.

Acute usage of these substances and other stimulatory behaviors induce feelings of well-being via neuronal DA release at the NAc. Prolonged abuse results in a toxic “pseudo” feeling of well-being, leading to discomfort, tolerance, and disease. Thus, a decreased number of DA receptors, due to carrying a genotype that causes hypodopaminergia, leads to excessive cravings for psychoactive drugs and non-substance addictive behaviors, whereas a sufficient DA receptors density results in low craving behavior and greater reward satisfaction. One goal to help prevent substance abuse would be to increase DA D2 receptors in genetically prone individuals. While in vivo experiments using a standard D2 receptor agonist induces downregulation of the receptor, experiments in vitro have demonstrated that constant stimulation of the DA receptor system via a known D2 agonist leads to a significant proliferation of D2 receptor coupled to G proteins despite any genetic antecedents. Thus, D2 receptor stimulation signals negative feedback mechanisms in the mesolimbic system to induce mRNA expression, causing the proliferation of D2 receptors [34].

The nutrigenomic activation of dopamine by the KB220-IV in a clinical trial using intravenous administration of the KB220 in more than 600 alcoholic patients resulted in significant reductions in RDS behaviors. The research hypothesis that manipulating the reward neural circuitry using amino–acid–enkephalinase therapy, oral and intravenous, would improve the behavioral and the emotional symptomology of 600 recovering alcoholics was an open trial clinical study. The results suggest that the combination of oral and intravenous administration of the KB220 variant, i.e., SG8839, significantly improved the behavioral and emotional recovery of the alcoholic subjects. The comparison of the pre, and post-administration scores included a reduction of depression (*p* < 0.001), anxiety (*p* < 0.001), fatigue (*p* < 0.001), anger (*p* < 0.001), lack of energy (*p* < 0.001), crisis (*p* < 0.001), and craving (*p* < 0.001). The mean reductions for depression (61.0 ± 6.3%), anxiety (53.8 ± 10.2%), craving (76.3 ± 3.1%), fatigue (76.9 ± 3.1%), and crisis (53.8 ± 5.5%) were all significantly greater than 50% (*p* < 0.001). This study that first combined oral and intravenous KB220 resulted in clinical improvement [37].

An expanded study of the oral KB220Z complex confirmed these results (7). Future studies must await both positron emission tomography (PET) scanning and functional magnetic resonance imaging (fMRI) to determine the effects of oral KB220Z on D2 receptor density and the status of reward and motivational brain regions (e.g., ventral striatum, amygdala, and orbitofrontal cortex). Confirmation of these results in large, population-based, and case-controlled experiments is necessary [34]. These studies would contribute important information that could eventually lead to significant improvement in recovery for those with dopamine deficiency-RDS due to the breakdown of multiple neurotransmitter signal transductions in the brain reward cascade [2], a factor in various types of addictions.

## 9. Quantitative Electroencephalogramo (qEEG) Studies

There are currently at least 45 studies showing a wide range of benefits with some RDS endophenotypes, see [17]. Based on clinical trials and animal research as presented herein, the pro-dopamine regulator, known in its original prototype form as KB220, shows promise in the areas of addiction and pain. Other genetic and neurobiological studies are required to elucidate the mechanism of action of this neuro-nutrient. The evidence to date points to the induction of “dopamine homeostasis” and an epigenetically induced normalization of neurotransmitter signaling and the associated asymptomatic restoration of brain reward cascade (BRC) function. As published over the last 50 years, these results encourage the continued development of appropriate nutrigenomic solutions for the millions of victims of all addictions, called reward surfeit syndrome (RSS) in adolescents and reward deficiency syndrome (RDS) in adulthood [38]. Quantitative electroencephalogram (qEEG) demonstrated activation of the mesolimbic system by a variant of KB220Z [39]. Positive findings using qEEG imaging in a randomized, triple-blind, placebo-controlled, crossover study of the oral KB220Z complex in abstinent subjects with psychostimulant use disorder showed increased alpha waves and decreased beta waves in the parietal brain region. Using t-statistics, significant differences observed between the placebo and the KB220Z complex consistently occurred in the frontal regions after week one and week two of the analyses (*p* = 0.03). This first report showed the prefrontal cortex’s involvement in the qEEG response to a natural putative D2 agonist (KB220Z), especially in subjects with the dopamine D2 A1 allele. More support for this finding comes from an additional study of 14 severe multi-drug abusers, who carried the DRD2 A1 allele, undergoing protracted abstinence. There were significant qEEG differences between those who received one dose of placebo and those administered the KB220Z. The KB220Z generated positive regulation of the dysregulated electrical activity of the brain in these addicts. The results indicate a phase change from low amplitude or power in the brain to a more functional state by increasing an average of 6.169 mV(2) across the prefrontal cortical region. The first experiment demonstrated that while 50% of the subjects carried the DRD2 A1 allele, 100% carried ≥ one risk allele. The proposed genetic addiction risk severity test for these 14 subjects revealed that 72% had moderate-to-severe addiction risk. Repeating the experiment in three additional currently abstinent polydrug abusers carrying the DRD2 A1 allele found similar results [40].

The authors have previously demonstrated for the first time that intravenous administration of the original KB220 (primarily amino acids and trivalent chromium without any of the saccharide-rich botanicals) reduces or “normalizes” aberrant electrophysiological parameters of the reward circuitry site. The published pilot study reported that administering one intravenous dose of KB220 significantly normalized the abnormal qEEGs of a heroin abuser and an alcoholic during protracted abstinence (widespread alpha and widespread theta activity, respectively). Both patients were genotyped for various neurotransmitter reward genes to determine their risk of developing heroin or alcohol dependence [39]. The genes tested included the dopamine D4 receptor exon 3 Variable Number Tandem Repeats (VNTR)S (DRD4), DRD2 TaqIA (rs1800497), the dopamine transporter (DAT1, locus symbol SLC6A3), monoamine oxidase A upstream VNTR (MAOA-uVNTR), serotonin transporter-linked polymorphic region (5HTTLPR, locus symbol SLC6A4), and COMT val158 met Single Nucleotide Polymorphisms (SNP) (rs4680) [39].

## 10. Functional Magnetic Resonant Imaging (fMRI) Study Evidence for Dopamine Homeostasis

The powerful effects of coupling GARS and KB220Z, as evidenced by healthy dopamine homeostasis seen in recent fMRI studies [18,19], have clearly shown the importance of pro-dopamine neuro-regulation. Firstly Febo et al. [19] showed that the pro-dopaminergic nutraceutical (KB220Z) augments significantly above placebo, functional connectivity between the cognitive and reward regions in rat brains. These areas include the nucleus accumbens, anterior cingulate gyrus, anterior thalamic nuclei, hippocampus, prelimbic and infralimbic loci. The nutraceutical KB220 significantly increased functional compartmental brain interconnectivity, ‘crosstalk,’ and volume recruitment, potentially neuroplasticity. Functional connectivity increases and dopaminergic functionality found across the reward circuitry were specific to the reward regions rather than being broadly distributed in the brain.

The robust yet selective response in drug naïve rodents has clinical relevance for recovering individuals at risk for relapse, who often show decreases in functional connectivity after protracted withdrawal. Additional studies will evaluate KB220Z in animal models of addiction. Blum et al. also found that the KB220Z significantly normalized reward circuitry neurotransmission within one hour of administration in 10 heroin addicts following abstinence sustained of an average of 16.9 months. Five subjects participated in a triple-blinded placebo-controlled randomized crossover study of KB220Z experiment. Triple-blinded experiments have the person administering the treatment, the person evaluating the response, and the subject blinded to placebo or treatment. Blum et al. found that KB220Z induced increased blood-oxygen-level dependence (BOLD) signaling and activation in caudate-accumbens-dopaminergic pathways compared to placebo after one-hour acute administration of the treatment.

Additionally, KB220Z reduced resting-state activity in the cerebellum of abstinent heroin addicts. For the second phase of this pilot study, three brain regions were significantly activated from a resting state by KB220Z compared to placebo treatment (*p* < 0.05) for all ten abstinent heroin-dependent subjects. A new network of observed enhanced functional connectivity included the medial frontal gyrus, dorsal anterior cingulate, nucleus accumbens, posterior cingulate, occipital cortical areas, and cerebellum. These fMRI study results suggest a putative anti-craving, anti-addiction relapse role of KB220Z by indirect or direct dopaminergic interaction. Since there was a small sample size, we caution against a definitive interpretation of these preliminary results. Confirmation with additional research and ongoing human and rodent studies of KB220Z is necessary.

## 11. Biphasic Approach to Addiction Treatment

One purpose of this article is to promote the development of novel IV compounds to induce “dopamine homeostasis” and provide new tools to assist clinicians and health professionals on the frontline [41,42,43,44,45,46,47,48,49,50,51,52,53,54,55,56,57,58,59]. While we understand the conventional perspective of reducing harm by utilizing, for example, opioid replacement therapy (ORT), we now suggest an evidence-based complementary approach. One option is to utilize ORT to address the problem in the short term. However, over longer periods, one could provide an evidence-based intervention to epigenetically repair either the trait (genetic) or state (epigenetically induced over at least two generations) neurochemistry of the brain reward circuit [39,60], particularly during the known opioid crisis, as eloquently discussed in Oesterle et al. [1].

While more research would be beneficial, let us at least initiate acceptable guidelines that include the understanding of RDS as an umbrella term for all addictive-type behaviors. Understanding the neurogenetics, as pointed out in the scientific community, and utilizing a more customized ‘systems biology’ approach (Precision Addiction Management) as outlined herein, seems most prudent and represents a step forward in the recovery process of the many millions of those afflicted with RDS globally [61]. A biphasic approach may be a good alternative: a short-term blockage followed by long-term dopaminergic upregulation. The treatment goal would be to augment brain reward functional connectivity volume and target the stress-like anti-reward symptomatology of addiction and reward deficiency. The GARS test indicates these phenotypes and dopamine homeostasis achieved via “Precision Addiction (or “Behavioral”) Management” (PAM or PBM): the customization of neuronutrient supplementation based on the GARS test result along with behavioral interventions.

## 12. Limitations and the Future

Dopaminergic epigenetic regenerative treatments are necessary to reverse genetic RDS and acquired syndromes that reduced quality of life and increase the likelihood of overdose, addiction, and suicide. Swenson et al. [62] suggested vigorous physical exercise as one dopaminergic regenerative treatment and transcranial magnetic stimulation (TMS) studied by Raij et al. [63] to treat post-addiction anhedonic states. These approaches are complementary and can be added as part of an addiction treatment recovery program.

Understanding that while there is evidence for this consequential approach of coupling GARS with a DNA guided precision nutrigenomic therapeutic to help induce “dopamine homeostasis, more clinical research is not only prudent but increasingly valuable. The purpose of this article is to encourage neurological and brain mapping animal and human addiction research to facilitate the potential movement from bench to bedside. There are many unanswered questions based on this pioneering work, which require additional neuroimaging studies. An example would be determining the possibility that KB220Z induces increases in mRNA expression of aberrant DNA polymorphisms for reward genes measured by GARS or other similar genetic testing panels. The potential of coupling DNA testing with mRNA profiling in primary, secondary, and tertiary treatment of RDS is a laudable goal for the future.

## 13. Summary

The contemporary literature confirms that an array of polymorphic genes related to neurotransmitters and secondary messengers control dopamine’s net release in the nucleus accumbens (NAc), which resides in the mesolimbic region of the brain [64,65,66,67,68,69,70,71,72]. They are linked primarily to motivation, anti-stress, incentive salience (wanting), metabolic and immune-incompetency, and well-being. The Nobel Prize was granted to Carlsson, Greengard, and Kandel in 2000 for their work on the cellular and molecular function of dopaminergic activity at neurons, including the memory and fear response. At this time, Americans are facing their second and worst opioid epidemic. Deaths due to overdose, emergency room visits, health consequences, and substance use disorders have increased, as have suicide deaths [73]. Abuse of psychoactive substances causes anhedonia and despair, which have only increased during this pandemic. Prescribed opioids for non-malignant pain, heroin access, and the emergence of cheap, potent synthetic opioids drive this epidemic of overdoses and OUDs [74]. Currently, the clinical consensus is to treat OUD as if it were an opioid deficiency syndrome with long-term to life-long opioid substitution therapy [75]. Using opioids to treat OUD forever seems counterintuitive at best. Due to the current opioid epidemic and the dismal rate of sustainable and prolonged recovery, the revolving door MAT treatment strategy of using opioid agonist, and antagonist therapy, and dopamine antagonists needs reconsideration [3].

However, for some patients, opioid agonist administration may be seen as necessary to replace missing opioids, treat OUD, prevent overdoses, and require lifetime use similar to insulin as a chronic therapeutic, especially if there are genetic antecedents [76]. Treatment of OUD and addiction is similar to the endocrinopathy conceptualization in that it views opioid agonist MAT as an essential core to therapy [77]. Knowing who has a deficiency syndrome may inform treatment as well as prevention efforts in the future.

We encourage clinicians to research and understand the importance of a molecular framework to explain the current underpinnings of endorphinergic/dopaminergic mechanisms related to opioid deficiency syndrome and a generalized reward processing deficiency [19,78,79,80,81,82,83,84,85,86,87,88,89,90,91,92,93,94,95,96,97]. Along these same lines, many RDS subtypes have also been linked to hypodopaminergia using sophisticated imaging techniques [98,99,100,101,102,103,104,105,106,107,108,109,110,111,112]. Can we better combat SUD through early genetic risk screening enable early intervention by the induction of dopamine homeostasis? Safe and effective dopamine agonist technologies that restore optimal gene expression and rebalance neurotransmitter interconnectivity in the brain reward cascade are currently viable options. The ideas reviewed here are summarized in Figure 2.

## Figures and Tables

**Figure 1 jpm-11-00212-f001:**
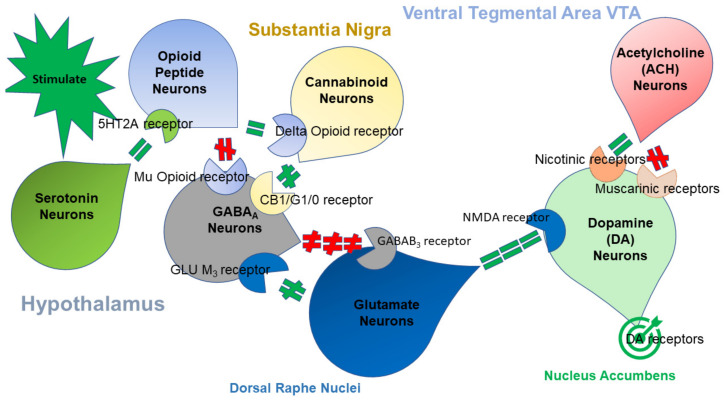
This figure illustrates the interaction of at least seven major neurotransmitter pathways implicated in the brain reward cascade (BRC). Within the hypothalamus, environmental stimulation causes serotonin release, which may activate 5HT-2a receptors (the green, equal sign). The opioid peptides have two distinct effects, possibly via two different opioid receptors. One is to inhibit (the red hash sign) through the mu-opioid receptor (potentially via enkephalin) and project to the Substantia nigra to Gamma-Aminobutyric acid A (GABAA) neurons. The second is to simultaneously project (the green, equal sign) to Cannabinoid neurons (e.g., Anandamide and 2-archydonoglcerol) through Beta–Endorphin link to delta receptors, which in turn inhibit GABAA neurons at the Substantia nigra. When cannabinoids (principally 2-archydonoglcerol) are activated, they can also indirectly disinhibit (the green hash sign) GABAA neurons in the Substantia nigra through activation of G1/0 coupled to CB1 receptors. Similarly, Glutamate neurons located in the Dorsal Raphe Nuclei (DRN) can indirectly disinhibit GABAA neurons in the Substantia Nigra by activating Glutamine (GLU). M3 receptors (the green hash sign). GABAA neurons, when stimulated, will powerfully (the red hash signs) inhibit Ventral Tegmental Area (VTA) glutaminergic drive via Gamma-Aminobutyric acid B GABAB 3 neurons. Finally, Glutamate neurons in the VTA will project to dopamine neurons through N-methyl-D-Aspartate (NMDA) receptors (the green, equal sign) to preferentially release dopamine at the Nucleus Accumbens (NAc) (shown as a bullseye), indicating good feeling (modified Blum et al. [2] with permission). Key: Activate—the green, equal sign; Inhibit—the red hash sign; Disinhibit—the green hash sign

**Figure 2 jpm-11-00212-f002:**
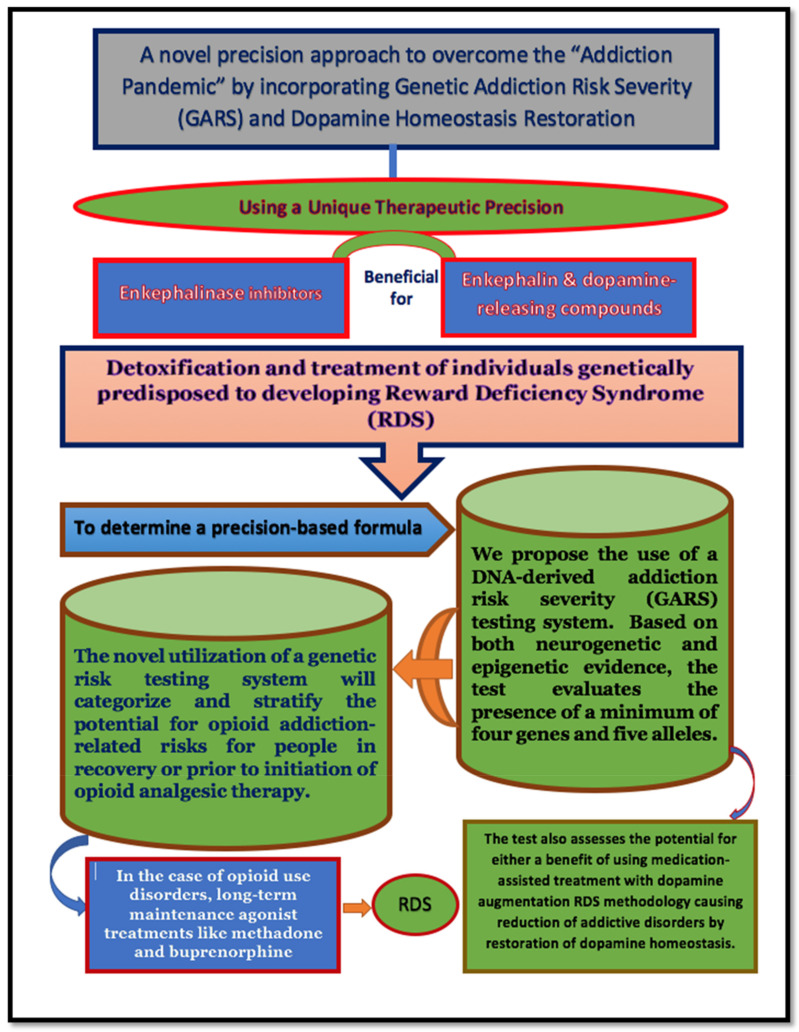
This figure is a graphic illustration of how the research and concepts explored here may be applied from bench to bedside.

**Table 1 jpm-11-00212-t001:** Summary of Studies used for Cases vs. Controls for Alcohol Use Disorder (AUD) Studies.

GENE	Risk Allele	Phenotype	# of Studies	Pat ** Case (N) Control (N)	Meta-Analysis *** (N)	Sig (*p*)	Comment
**DRD1**	Rs4532 & specific haplotype rs686*T-rs4532*G within the DRD1 gene	Alcohol Use Disorder (AUD) and Aggression & Impulsivity	Three	Case (569) Cont (218)	NONE	<0.1–0.01	POSITIVE for Alcohol Dependence and related phenotypes like aggression & Impulsivity
**DRD2**	Rs1800497	Severe alcoholism, long-term drinking, alcohol dependence, parental rule-setting, comparison severe vs. less severe alcoholics, relapse and ASI after 12 years in 12 step programs, Family linkage, heavy drinking, early-onset, stress, harm avoidance and antisocial behavior related to AUD, severe medical consequences, mortality hospitalization, Children of Alcoholics (C.O.A.s), parental history of alcoholism, & drinking in the general population,	Sixty–Two	Case (17,382) Cont (17,036)	Four (4)	<0.04–0.09	POSITIVE for Alcohol Dependence and related phenotypes
**DRD3**	DRD3 Ser9Gly polymorphism (rs6280)	Alcohol Dependence (AD), Anhedonia and Major –depressive disorder and Obsessive-Compulsive Drinking	Three	Case (545) Cont (156)	NONE	<0.001–0.008	POSITIVE for Alcohol Dependence (AD), Anhedonia and Major –depressive disorder and Obsessive-Compulsive Drinking
**DRD4**	Rs180095 48bP repeat VNTR	Risk Factor for Alcoholism, Alcohol Dependence Smoking Behavior, Polysubstance abuse, higher rates of novelty seeking, higher lifetime alcoholism, generalized addiction, increased influence of peer pressure to drink, problematic alcohol use, increase the risk for severity of alcoholism, blunted response to alcohol cues, increase in alcohol craving, increased risk for social bonding with fellow alcoholics.	Forty –Eight	Case (11,740) Cont (9365)	Two (2)	<0.06–0.05	POSITIVE for many alcohol-related phenotypes
**DAT1**	9R allele compared to 10R.	Alcoholism, alcohol consumption, alcohol withdrawal symptoms (AWS) and delirium tremens (DT), number of drinking days, vulnerability to alcoholism, and families with alcoholism compared to families without alcoholism	Twenty–Four	Case (4644) Cont (3761)	Two (2)	<0.05–0.09	POSITIVE for Alcoholism, alcohol consumption, alcohol withdrawal symptoms (AWS) and delirium tremens (DT), number of drinking days, vulnerability to alcoholism, and families with alcoholism compared to families without alcoholism
**COMT**	Rs4680 Catechol-O-methyl-transferase (COMT) Val158Met	Alcohol Dependence (AD), alcohol intake past year, generalized SUD, Alcohol & Tobacco consumption, drug abuse, in alcoholics reduced dopamine receptor sensitivity	Seventy–Five	Case (10,018) Cont (8861)	One (1)	<0.01–0.01	POSITIVE for Alcohol Dependence (AD), alcohol intake past year, generalized SUD, Alcohol & Tobacco consumption, drug abuse, in alcoholics reduced dopamine receptor sensitivity
**OPRM1**	OPRMI (rs1799971)	Alcohol Dependence (AD), Severity of AWS, sensitivity to dopamine receptors, alcohol consumption, depression, response to alcohol cues and relapse risk, alcohol sensitivity in adolescents, drinking frequency, vulnerability for alcohol to hijack the reward system, alcohol craving, alcohol-related hospital readmission, more readmissions, and fewer days until the first readmission	Fifteen	Case (6428) Cont (5196)	One (1)	<0.047–0.06	POSITIVE for Alcohol Dependence (AD), Severity of AWS, sensitivity to dopamine receptors, alcohol consumption, depression, response to alcohol cues and relapse risk, alcohol sensitivity in adolescents, drinking frequency, vulnerability for alcohol to hijack the reward system, alcohol craving, alcohol-related hospital readmission, more readmissions, and fewer days until the first readmission
**GABRB3**	Receptor beta3 subunit (GABRB3) 181 variant	The risk for Alcoholism, the onset of drug abuse in Children of Alcoholics (COAS), Parental transmission and alcoholism, hypodopaminergia, Mood-related alcohol expectancy (AE), drinking refusal self-efficacy (DRSE), depression, and prevalence in COAS	Four	Case (196) Cont ()	NONE	<0.05–0.07	POSITIVE for risk for Alcoholism, the onset of drug abuse in COAS, Parental transmission and alcoholism, hypodopaminergia, Mood-related alcohol expectancy (AE), drinking refusal self-efficacy (DRSE), depression, and prevalence in COAS
**MAOA**	30 BP. VNTR-3.5R, 4R DN repeat polymorphisms	Alcohol Dependence, impulsivity, antisocial personality, susceptibility to alcoholism, smoking behavior, poor psychosocial environment, and lower age of onset of alcoholism.	Five	Case (731) Cont (1111)	NONE	<0.043–0	POSITIVE for Alcohol Dependence, impulsivity, antisocial personality, susceptibility to alcoholism, smoking behavior, poor psychosocial environment, and lower age of onset of alcoholism
**SLC6A4** **(5HTTLPR)**	promoter region (5-HTTLPR) (rs25531)	Alcohol Dependence, anxiety, age of onset, cue craving, lower socialization, depression, & polydrug abuse	Twenty–Seven	Case (13,328) Cont (2982)	Two (2)	<0.03–0.001	Alcohol Dependence, anxiety, age of onset, cue craving, lower socialization, depression, & polydrug abuse
TOTAL	NA	NA	268	Case 65,581 Cont 48,686	Ten (10)	<0.06–0.009	

## Data Availability

Not Applicable.

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
