# Peer review of "A Novel Precision Approach to Overcome the “Addiction Pandemic” by Incorporating Genetic Addiction Risk Severity (GARS) and Dopamine Homeostasis Restoration"

_jpm, 2021, doi:10.3390/jpm11030212_

Round 1

Reviewer 1 Report

In this review article, the authors present a systematic and comprehensive literature surveying on Reward Deficiency Syndrome and on the importance of dopamine homeostasis. Overall, the manuscript is clearly written and topically relevant. I feel that the manuscript is a strong candidate for publication in Journal of Personalized Medicine, but have a number of minor comments that should be addressed prior to publication.

Comments:

(1) Line 262~265 and Line 291, it is not clear to me what dopamine receptor proliferation means. Do the authors refer to increased coupling to G proteins?

(2) Line 283-284, “Thus, a decreased number of DA receptors, due to carrying a genotype that causes hypodopaminergia, leads to excessive cravings for psychoactive drugs and non-substance addictive behaviors.” It would be helpful to name a few genetic variants that cause reduced number of DA receptors, such as DRD2 A1 allelic genotype.

(3) References for the statement in Line 296~Line 309 are lacking.

Author Response

(1) Line 262~265 and Line 291, it is not clear to me what dopamine receptor proliferation means.

“Studies in vitro have shown that constant stimulation of DA receptors by agonists result in proliferation of Dopamine D2 receptors.” From Ref [32].

Based on our qEEG studies and previous research [33], we cautiously state that long-term activation of dopaminergic receptors (i.e., DRD2 receptors) will give rise to a greater proliferation of the receptors and result in enhanced “dopamine sensitivity.” Dopamine receptor proliferation can lead to an increased sense of happiness, particularly in carriers of the DRD2 A1 allele [32].

Do the authors refer to increased coupling to G proteins?

Yes, due to increased number of D2 receptors as Boundy discovered [33]. Green Highlight added text and reference:

  1. Boundy, V.A.; Lu, L.; Molinoff, P.B. Differential coupling of rat D2 dopamine receptor isoforms expressed in Spodoptera frugiperda insect cells. Journal of Pharmacology and Experimental Therapeutics 1996, 276, 784.

While in vivo experiments using a standard D2 receptor agonist induces downregulation of the receptor, experiments in vitro have demonstrated that constant stimulation of the DA receptor system via a known D2 agonist leads to a significant proliferation of D2 receptor coupled to G proteins despite any genetic antecedents. Thus, D2 receptor stimulation signals negative feedback mechanisms in the mesolimbic system to induce mRNA expression, causing the proliferation of D2 receptors [32].  

(2) Line 283-284, “Thus, a decreased number of DA receptors, due to carrying a genotype that causes hypodopaminergia, leads to excessive cravings for psychoactive drugs and non-substance addictive behaviors.” It would be helpful to name a few genetic variants that cause reduced number of DA receptors, such as DRD2 A1 allelic genotype.

An example of a genetic allele that reduces dopaminergic function is the Val allele of the COMT gene. Additional text is highlighted in green.

274-279 “Examples are those who possess a paucity of dopaminergic and serotonergic receptors or an increased rate of synaptic DA catabolism due to expression of the high catabolic genotype from the COMT gene are predisposed to self-medicating via any substance or behavior such as alcohol, nicotine, psychostimulants, opiates, sex, excessive eating, gambling, and gaming, that will activate DA release.” The high catabolic genotype from the COMT gene is the Val allele functions to catabolize dopamine in synapse.

(3) References for the statement in Line 296~Line 309 are lacking.

The nutrigenomic activation of dopamine by the KB220-IV in a clinical trial using intravenous administration of the KB220 in more than 600 alcoholic patients resulted in significant reductions in RDS behaviors. The research hypothesis that manipulating the reward neural circuitry using amino–acid–enkephalinase therapy, oral and intravenous, would improve the behavioral and the emotional symptomology of 600 recovering  alcoholics was an open trial clinical study. The results suggest that the combination of oral and intravenous administration of the KB220 variant, i.e., SG8839, significantly improved the behavioral and emotional recovery of the alcoholic subjects. The comparison of the pre, and post-administration scores included a reduction of depression (p<0.001), anxiety (p<0.001), fatigue (p<0.001), anger (p<0.001), lack of energy (p<0.001), crisis (p<0.001), and craving (p<0.001). The mean reductions for depression (61.0±6.3%), anxiety (53.8±10.2%), craving (76.3±3.1%), fatigue (76.9±3.1%), and crisis (53.8±5.5%) were all significantly greater than 50% (p<0.001). This study first combined oral and intravenous KB220 resulted in clinical improvement [34].

34           Blum K; Chen TH; Downs BW; Meshkin B; Blum SH; Martinez Pons  M; Mengucci JF; Waite RL; Arcuri V; Varshofsiky M, et al. Synaptamine (SG8839),™ An Amino-Acid Enkephalinase Inhibition Nutraceutical Improves Recovery of Alcoholics, A Subtype of Reward Deficiency Syndrome (RDS).

Reviewer 2 Report

In the review “A novel precision approach to overcome the “Addiction Pandemic” by incorporating Genetic Addiction Risk Severity (GARS) and Dopamine Homeostasis Restoration” Blum et al has proposed their views on treatment of individuals genetically predisposed to developing reward deficiency syndrome (RDS). Review is nicely written and informative. I don’t have any specific comment; but few suggestions which might make the review interesting to a larger audience.

  1. In the initial part of the review authors has explained RDS. There are no next generation large scale genomics studies on RDS till date. Within section “Genetic polymorphisms of RDS: Case-control studies, for alcoholism” author can send a message to new generation researcher to investigate the molecular mechanism behind RDS using next generation genomics approach.
  2. While discussing RDS author should discuss the animal model study if available.
  3. There are some issues with structure of the review. Initial sections were arranged with 1, 1.2, 2 --- and then suddenly it disappear. I will be little hard for the readers to follow in current format.

Author Response

  1. In the initial part of the review authors has explained RDS. There are no next-generation large-scale genomics studies on RDS till date.

Within section “Genetic polymorphisms of RDS: Case-control studies, for alcoholism” author can send a message to new generation researcher to investigate the molecular mechanism behind RDS using next-generation genomics approach. 

234- Genetic polymorphisms of RDS: Case-control studies, for alcoholism

Next-generation large-scale genomics studies have had limited success in identifying alleles associated with addiction and RDS. Although GWAS and Next Generation Sequencing are now available as important genetic tools, there are some fundamental issues. Certainly, GWAS, for example, is useful to identify new clusters of genes that may relate to an etiological factor as a genetic antecedent to specific RDS behaviors like chemical dependency, the next important step following GWAS results is subsequent convergence to specific candidate genes. Thus, if there is indeed a blueprint or clue as to a specific known gene and associated polymorphic risk allele to link to a specific phenotype such as SUD or even Cannabis Use Disorder, although the contribution of each gene may be small, it is still significant.  

Being cognizant of these difficulties and awaiting further research, the BRC was utilized as a blueprint for a literature review to determine each gene and associated polymorphism proposed in the GARS panel in case-control studies; the studies listed in table 1 are specifically for alcohol use disorder (AUD).

  1. While discussing RDS author should discuss the animal model study if available.

179-182. Drug, food, or behavioral addictions characterized by reduced dopaminergic function linked earlier to the DRD2 gene A1 allele (1), and now an array of reward gene alleles manifest clinically as Reward Deficiency Syndrome (RDS) [23]. Indeed, animal models of RDS are available [23]; in fact, Willuhn et al. [24] showed that there is a phasic dopamine deficiency as seen in RDS when animals increase lever pressing for more cocaine. There is also evidence for the potential utility of a natural, non-addictive, and safe putative D2 agonist that can treat patients in recovery from Reward Deficiency Syndrome (RDS), including psychoactive substance use disorder (SUD) [24].

  1. There are some issues with structure of the review. Initial sections were arranged with 1, 1.2, 2 --- and then suddenly it disappear. I will be little hard for the readers to follow in current format.

Thank you for the alert. Have renumbered the sections.

Reviewer 3 Report

For ethical reasons, I do not feel comfortable performing a proper review of the manuscript by Bloom K and collaborators. In my opinion, there is a major discrepancy between the title of the article and the content of the manuscript. The title suggests that the topic of the review is about how Genetic Addiction Risk Severity (GARS) could be a trick to prevent the “addiction pandemic”. The content of the manuscript seems rather different, describing the potential beneficial effects of KB220 and KB220Z, two probiotics invented by the first author of the review, to prevent such pandemic. All studies indicating beneficial effects of these two probiotics are self-citations, which points to a lack of objectivity of the authors.

Moreover, several major issues need to be addressed before any potential publication:

  • Addiction is a complex disease, involving many neurotransmitters and brain regions. It is not only a matter of dopamine homeostasis and dopamine D2 gene expression or gene allele. As an example, opioid, cannabinoid and serotonigergic systems within the nucleus accumbens or the VTA play major role in the modulation of the dopaminergic neurons. In this framework, Figure 1 appears rather minimalist, mentioning only part of these systems in each brain regions where they are present, and needs major improvement.
  • Line 179: The cascade of neurotransmission the blueprint (?): “The Brain Reward Cascade (BRC) schematic depicts the known interactions of at least seven neurotransmitter-pathways involved in the neurotransmission of reward. The complexity of the interactions within the BRC demand frequent revision of this schematic (see Figure 1)”. I totally agree with the authors, which does not mean that the authors should not develop properly this section, to allow the readers picturing how complex such cascade can be.
  • The authors should also improve the graphical quality of Figure 2. Moreover, I do not understand how this figure fits in the section “limitation & future”. Authors should clarify this.
  • Authors ought to mention if the studies cited in their manuscript were performed in humans or in animals (rodents).
  • Line 175: “There is also evidence for the potential utility of a natural, non-addictive, and safe putative D2 agonist that can treat patients in recovery from Reward Deficiency Syndrome (RDS), including psychoactive substance use disorder (SUD) [24].” The authors mentioned a review in which no experimental evidence is provided whatsoever that would confirm such hypothesis.
  • Line 82: “During the current viral pandemic…” I do not understand the logical link between the viral pandemic and opioid use disorder as discussed later in the review. Please remove.
  • In this review, at least 31 citations among 107 are self-citations, which seems rather excessive to me.

Author Response

For ethical reasons, I do not feel comfortable performing a proper review of the manuscript by Bloom K and collaborators. In my opinion, there is a major discrepancy between the title of the article and the content of the manuscript.

The title suggests that the topic of the review is about how Genetic Addiction Risk Severity (GARS) could be a trick to prevent the “addiction pandemic.”

Yes, it is about a trick or “approach to treating opioid addiction.” The treatment approach is based on many years of basic and theoretical science; the earliest peer-reviewed research paper cited here is O’Hollaren P, 1961. Articles about the GARS test developed by Blum’s group have been published in many peer-reviewed journals. A PUBMED search now has 58 articles listed.  

The content of the manuscript seems rather different, describing the potential beneficial effects of KB220 and KB220Z, two probiotics invented by the first author of the review, to prevent such pandemic.

KB220 and KB220Z are Research Identification Codes used to research various neuro nutrient formulations developed by Blum et al. to treat RDS, including addictions.

The WHO defines a probiotic as “live micro-organisms which, when administered in adequate amounts, confer a health benefit on the host.” This is the internationally endorsed definition of probiotics. KB220 is not a probiotic.

All studies indicating beneficial effects of these two probiotics are self-citations, which points to a lack of objectivity of the authors.

The developer Prof Kenneth Blum has been involved in all research carried out on KB220 as he is supplying the formulation; however, the basic research that lead to the theoretical basis of the formulation was carried out over many years in many laboratories. Also, the research on KB220 and KB220Z are both clinical and experimental, some involving crossover triple blinded fMRI.

Moreover, several major issues need to be addressed before any potential publication:

  1. Addiction is a complex disease, involving many neurotransmitters and brain regions. It is not only a matter of dopamine homeostasis and dopamine D2 gene expression or gene allele. As an example, opioid, cannabinoid and serotonergic systems within the nucleus accumbens or the VTA play major role in the modulation of the dopaminergic neurons. In this framework, Figure 1 appears rather minimalist, mentioning only part of these systems in each brain regions where they are present, and needs major improvement.

Please help us to improve on Figure 1. This figure praised by many in the field, is i a teaching schematic that has been modified as research using imaging techniques have identified neurotransmitter pathways.

  1. Line 179: The cascade of neurotransmission the blueprint (?):

Yes, the BRC has been used by Prof. Blum as a blueprint initially for the choice of genes in the research that lead to the discovery of the association of the DRD2A1 allele that with severe AUD in 1990, and subsequently the formulation of KB220 and the selection of the alleles GARS test, apropos here.

“The Brain Reward Cascade (BRC) schematic depicts the known interactions of at least seven neurotransmitter-pathways involved in the neurotransmission of reward. The complexity of the interactions within the BRC demand frequent revision of this schematic (see Figure 1)”.

I totally agree with the authors, …

Thank you.

…which does not mean that the authors should not develop properly this section, to allow the readers picturing how complex such cascade can be.

Several of the referenced papers do this [2,4,18,40, 89, 98].

The authors should also improve the graphical quality of Figure 2. Moreover, I do not understand how this figure fits in the section “limitation & future”. Authors should clarify this.

Figure moved to “Summary,” Thank you, good catch.

Authors ought to mention if the studies cited in their manuscript were performed in humans or in animals (rodents).

They do, rat and rodent studies are identified, However, to eliminate any confusion about the qEEG studies, the green highlighted text was added.

Line 362-366. Quantitative electroencephalography (qEEG) demonstrated activation of the mesolimbic system by a variant of KB220Z [44]. Positive findings using qEEG imaging in a randomized, triple-blind, placebo-controlled, crossover study of the oral KB220Z complex in abstinent subjects with psychostimulant use disorder showed increased alpha waves and decreased beta waves in the parietal brain region.

Line 175: “There is also evidence for the potential utility of a natural, non-addictive, and safe putative D2 agonist that can treat patients in recovery from Reward Deficiency Syndrome (RDS), including psychoactive substance use disorder (SUD) [24].”

The authors mentioned a review in which no experimental evidence is provided whatsoever that would confirm such hypothesis.

The paper is a “Reconsideration of data derived from animal studies.” As such, this comment is a non sequitur and does not reflect the reviewer’s comment.

Line 82: “During the current viral pandemic…” I do not understand the logical link between the viral pandemic and opioid use disorder as discussed later in the review. Please remove.

No, the point is that the viral pandemic has exacerbated the SUD epidemic with an increase of up to 40% overdoses.

In this review, at least 31 citations among 107 are self-citations, which seems rather excessive to me.

The first Blum et al. GARS research article mentioned in this Invited Review was published in1990. Over that 30 years of research, an average of about two papers per year. Not so excessive. Dr. Blum is not trying to self -cite; instead, he imparts data relevant to the paper’s subject matter.